# Immunomodulatory drug fingolimod (FTY720) restricts the growth of opportunistic yeast *Candida albicans in vitro* and in a mouse candidiasis model

Niloofar Najarzadegan[1], Mahboobeh Madani[1]*, Masoud Etemadifar[2], Nahad Sedaghat[3,4]

1 Department of Microbiology, Falavarjan Branch, Islamic Azad University, Isfahan, Iran, 2 Department of Neurosurgery, School of Medicine, Isfahan University of Medical Sciences, Isfahan, Iran, 3 Alzahra Research Institute, Alzahra University Hospital, Isfahan University of Medical Sciences, Isfahan, Iran, 4 Network of Immunity in Infection, Malignancy and Autoimmunity (NIIMA), Universal Scientific, Education, and Research Network (USERN), Isfahan, Iran

* mmadani66@gmail.com

**Data Availability Statement:** All relevant data are within the paper and its Supporting Information files.

## Abstract

Fingolimod (FTY720) is a drug derived from the fungicidal compound myriocin. As it was unclear whether FTY720 has antifungal effects as well, we aimed to characterize its effect on *Candida albicans in vitro* and in a mouse candidiasis model. First, antifungal susceptibility testing was performed *in vitro*. Then, a randomized, six-arm, parallel, open-label trial was conducted on 48 mice receiving oral FTY720 (0.3 mg/kg/day), intraperitoneal *C. albicans* inoculation, or placebo with different combinations and chorological patterns. The outcome measures of the trial included serum concentrations of interleukin-10 and interferon-gamma, absolute lymphocyte counts, and fungal burden values in the mice's livers, kidneys, and vaginas. Broth microdilution assay revealed FTY720's minimum inhibitory concentration ($MIC_{99}$) to be 0.25 mg/mL for *C. albicans*. The infected mice treated with FTY720 showed lower fungal burden values than the ones not treated with FTY720 ($p<0.05$). As expected, the mice treated with FTY720 showed a less-inflammatory immune profile compared to the ones not treated with FTY720. We hypothesize that FTY720 synergizes the host's innate immune functions by inducing the production of reactive oxygen species. Further studies are warranted to unveil the mechanistic explanations of our observations and clarify further aspects of repurposing FTY720 for clinical antifungal usage.

## 1. Introduction

Fingolimod (FTY720) is an immunomodulatory drug currently indicated and approved by the United States Food and Drug Administration (FDA) for treatment of the relapsing forms of multiple sclerosis–an autoimmune entity involving the central nervous system. It is used off-label for progressive MS and other autoimmune neuropathies, while being evaluated for extra-nervous pathologies as well [1–3]. FTY720 is considered a prodrug of its phosphorylated

**Funding:** The author(s) received no specific funding for this work.

**Competing interests:** The authors have declared that no competing interests exist.

form fingolimod-phosphate (FTY720-P). Being a modulator of the sphingosine-1-phosphate receptors (S1PR) by structural analogy, FTY720-P induces internalization of S1PR from the surface of lymphocytes, hindering their trafficking outside the primary lymphatics and, therefore, keeping them away from inflammatory sites in end organs [4].

While being associated with beneficial outcomes and reasonable tolerability among people with immune-mediated conditions such as MS [5, 6], safety and toxicity studies have shown both dose-dependent and time-dependent adverse effects associated with FTY720 [7]. The time-dependent adverse effects have been opportunistic infections, reactivation of latent viruses, malignancies, etc.; these are mostly deemed to be due to chronic immunosuppression [7]. Among the dose-dependent adverse effects, the most common has been cardiotoxicity [5, 6]. S1PRs are highly expressed on cardiomyocytes; the initial agonistic effect of FTY720-P on S1PRs before inducing their internalization seems to be responsible for its dose-dependent cardiotoxic effects [8]. Hepatotoxic dose-dependent effects have also been reported [5–7], deemed to be associated with FTY720's metabolic burden on the liver [9]. Other dose-dependent unwanted and toxic effects of FTY720 due to S1PR agonism, e.g., macular edema [10], unwanted loss of weight and appetite [11], etc. have been reported.

FTY720 was derived from myriocin (ISP-1), which is a secondary metabolite of the entomopathogenic fungus *Isaria sinclairii* and a potent inhibitor of the serine palmitoyltransferase (SPT) enzyme in the *de novo* sphingolipid biosynthesis pathway [12]. Myriocin alters a metabolic pathway in eukaryotic cells vital for their stability and proliferation. Consistently, it has shown to have antifungal effects e.g., against *Candida* and *Aspergillus spp.* [13, 14].

While FTY720 has no activity against the SPT, emerging studies [3, 15] are pointing towards S1PR-independent anti-proliferative effects of FTY720 in its non-phosphorylated form. Meanwhile, several cases of opportunistic fungal infections are reported among people receiving FTY720 chronically [16, 17]. Although these reports describe case studies and population-based evidence is lacking in this regard, it seems unclear whether FTY720 facilitates fungal expansion *in vivo* by weakening the immune system, or has retained the antifungal effect of its molecular ancestor.

The subject of identifying and confirming the possible antifungal properties of substances like FTY720 –which could be administered safely in humans–gains more relevance when considering the rapid emergence of serious mycosis cases resistant to our usual antifungal armamentarium. Among the candidiasis cases, resistance to drugs is classically attributed to the less prevalent infections with non-albicans species, however, recent studies are documenting an alarming increase in the antifungal resistance of *Candida albicans*–the dominant pathogen in candidiasis cases [18, 19]. Considering that the incidence of opportunistic candidiasis is rising, and that commensal *C. albicans* has a high prevalence [20], its resistance to drugs could be a major complicating factor for the future candidiasis cases.

Hence, in order to address our gaps of knowledge, we first aimed to determine whether FTY720 has antifungal effects *in vitro* using antifungal susceptibility testing. Then, we aimed to characterize FTY720's effects on the immunological profiles, and on the fungal burden values in mice infected with *C. albicans*. We hereby report our studies in accordance with the Animal Research: Reporting of In Vivo Experiments 2.0 (ARRIVE 2.0) guidelines (available at: https://arriveguidelines.org/).

## 2. Methods

### 2.1 Design

The present studies were conducted from October 2019 until July 2020 in the Islamic Azad University of Falavarjan, Isfahan, Iran. As mentioned, a study was done *in vitro* to measure

FTY720's minimum inhibitory concentration (MIC) for *C. albicans*, and a study was done *in vivo* to characterize FTY720's effect on the immunological profiles and fungal burden values in murine candidiasis models. The later study (from now on referred to as the *in vivo* study) was designed with the following formulation of research question:

- Population: Female C57BL/6 mice.

- Interventions:

  ○ Oral FTY720

  ○ Intraperitoneal (IP) *C. albicans* inoculation

- Comparison:

  ○ Oral normal saline (oral placebo)

  ○ IP normal saline (injectable placebo)

- Outcomes and rationale of each:

  ○ Complete blood count with differential (CBC diff) and serum concentrations of interferon-gamma (IFNγ) and interleukin-10 (IL10) after 21 days of receiving oral intervention: to characterize the immunological profiles of the mice.

  ○ Colony count of *C. albicans* in the vaginas, kidneys, and livers of the mice after 21 days of receiving oral intervention: to measure the fungal burden [21].

Based on the formulation of the research question, the *in vivo* study was conducted with six arms (Fig 1):

1. Control: Six mice receiving oral and injectable placebos.

2. Only FTY720: Six mice, each treated with oral FTY720 and injectable placebo.

3. Only *C. albicans*: Six mice, each treated with oral placebo and an IP *C. albicans* inoculation.

4. FTY720 after *C. albicans*: Ten mice, each treated with oral FTY720 starting from two days after receiving an IP *C. albicans* inoculation.

5. FTY720 with *C. albicans*: Ten mice, each treated with oral FTY720 starting from the same day as receiving an IP *C. albicans* inoculation.

6. FTY720 before *C. albicans*: Ten mice, each treated with oral FTY720 starting from two days before receiving an IP *C. albicans* inoculations.

The only exclusion criterion for the *in vivo* study was death due to an unrelated reason. No other eligibility criteria were considered.

## 2.2 *In vitro* antifungal susceptibility testing

**2.2.1 Well diffusion assay.** We first used a *qualitative* well diffusion assay in order to i) estimate the probable *in vitro* MIC of FTY720 for *C. albicans* and ii) to have an estimation of the serial drug concentration measures required for the subsequent broth microdilution assay. For the well diffusion assay, 0.5 McFarland standard (1–5 x $10^6$ colony forming units (CFU)/ mL) preparations of *C. albicans* (CBS2747, Tehran University, Iran) were spread on three plates of Sabouraud dextrose agar (SDA) (Scharlau, Turkey). A total of eleven sample wells (four wells in two plates, and three wells in one plate), each with a diameter of six millimeters,

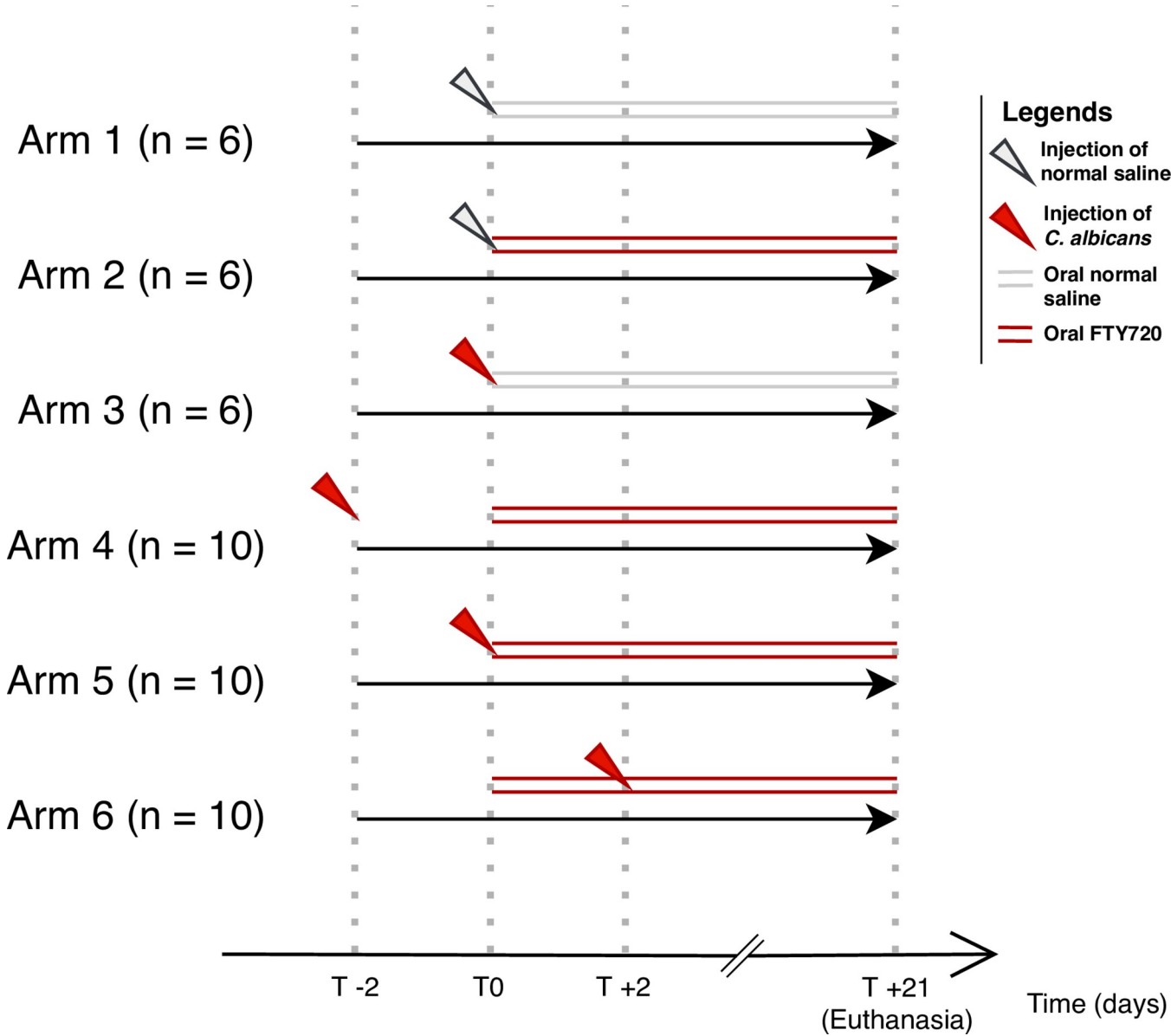

**Fig 1. Schematic overview of the parallel intervention arms of the *in vivo* study.**

were created in the plates using a sterile short (2 mL) Pasteur pipette. Then, using a standard sampler, one well was filled with normal saline as control and the others were filled with 200 μL suspensions of FTY720 with serial concentrations from 0.005 to 1.5 mg/mL. The presence or absence of a zone of inhibition around each well was determined without visual aid, after a 30-hour incubation of the plates at 37°C. The lowest concentration for which a zone of inhibition surrounded the corresponding well was considered as the probable MIC.

**2.2.2 Broth microdilution assay.** After obtaining the probable MIC using the well diffusion assay, a broth microdilution assay was performed following the methods described by the European Committee on Antimicrobial Susceptibility Testing (EUCAST) [22]. Briefly, microtubes in a sterile microtube plate were filled with 10 μL inocula of *C. albicans* (0.5 McFarland standard; 1–5 x $10^6$ CFU/mL), 90 μL of Sabouraud dextrose broth (SDB) (Scharlau, Turkey),

and 100 μL of serially-diluted suspensions of FTY720, from 0.0005 to 0.5 mg/mL (i.e., the microtubes finally contained $0.5–2.5 \times 10^5$ CFU/mL of *C. albicans* in SDB with serial dilutions of FTY720 from 0.00025 to 0.25 mg/mL). One microtube was added by normal saline instead of FTY720 (positive control), and one was filled with sterile SDB without the *C. albicans* inoculum (negative control). The optical density (OD) of each microtube was read and documented at 640 nm immediately after preparation, and after a 24-hour incubation at 37˚C. The minimum drug concentrations for which the post-incubation change in OD was <1% and <50% were considered as the $MIC_{99}$ and the $MIC_{50}$, respectively.

## 2.3 Preparation, allocation, and follow-up of the mice

For the *in vivo* study, 48 female C57BL/6 mice (Royan Institute, Iran), each weighing 22 (±3) grams, were prepared in accordance with the National Research Council guidelines for laboratory animal care [23]. From one week before randomization, all mice were kept in standard cages in a humidity-controlled room at a 25˚C (±2) temperature and a 12 hours daylight, 12 hours darkness cycle. Thereafter, each mouse was identification-marked by an ear tag, and allocated to each one of the study arms based on a random sequence generated by the NumPy software package (for Python version 2.7 on MacOS). They received their interventions in an open-label manner and were followed up daily until the endpoint of the study. All mice were treated with 100 μL liquid preparations of FTY720 (0.3 mg/kg/day) or normal saline, administered daily by gastric gavage for 21 days. This particular regimen of FTY720 was selected as it has been frequently used in previous studies, proven to bear immunomodulatory effects, and associated with minimal adverse reactions in C57BL/6 mice [24–30]. The mice assigned to be infected with *C. albicans* were inoculated with *C. albicans* through IP injections, with each inoculum containing $1.5 \times 10^5$ CFU of *C. albicans*; the other mice were injected with the same amounts of normal saline. At the end of the follow-up, the mice were sedated with ketamine-xylazine (100–10 mg/kg) and 1 mL of blood was taken from each through cardiac catheterization. The blood samples were taken in anticoagulated tubes, and were promptly sent to be prepared for CBC diff and cytokine assays. Then, through sterile procedures, the mice's livers, kidneys, and vaginal samples were extracted and sent to be prepared for fungal culture. The mice were finally euthanized with anesthetic overdose per American Veterinary Medical Association (AVMA) guidelines [31].

## 2.4 Analyses of samples

In order to characterize the mice's immune responses to the fungal infection, absolute WBC and lymphocyte counts, serum IL10 (an anti-inflammatory cytokine), and IFNγ (a proinflammatory cytokine) concentrations were measured. CBC diff was performed on whole blood samples manually (i.e., using a Neubauer chamber) [32]. IFNγ and IL10 concentrations were quantified using an enzyme-linked immunosorbent assay (ELISA) (BioAssay™ kit for mouse, USBiological, USA) in accordance with the manufacturer's instructions. For fungal culture, the vaginal samples were directly spread on SDA plates. The livers and kidneys were first homogenized and filtered (using sterile techniques and equipment), were diluted in ten serial concentrations, and then spread on SDA plates. After a 30-hour incubation at 37˚C, the numbers of *C. albicans* colonies were counted and documented.

## 2.5 Statistical methods

**2.5.1 Data preparation and determination of appropriate tests.** The distributions of immune response measures (IFNγ, IL10. WBC, and ALC) were assumed to be lognormal; these measures were $\log_{10}$-transformed before the analyses. The distributions of fungal burden

values were assumed to be normal. The normality/lognormality assumptions were tested using the Kolmogorov-Smirnov method [33, 34]; if the assumptions were confirmed by alpha>0.05, parametric tests (e.g., one-way analysis of variance [ANOVA]), and if not, non-parametric tests (e.g., Kruskal-Wallis) were used for those measures. Post hoc analysis and correction for multiple comparisons were performed by controlling the false discovery rate, using the two-stage step-up method of Benjamini, Krieger, and Yakutielli [35].

In order to determine whether a single pooled variance could be used in the parametric comparisons, the group variances were compared using the Brown-Forsyth test [36]. In cases of insignificant (*p*>0.05) difference among variances of the arms, a single pooled variance was calculated and used; otherwise, the Welch and Brown-Forsyth method [37] was used (variances were calculated for individual arms separately).

**2.5.2 Comparisons and their rationale.** To confirm the sufficiency of the follow-up period for FTY720 to implement its immunological effects, the IFNγ, IL10, WBC, and ALC measures were compared between the control mice (arm 1) and the mice receiving FTY720 with placebo injection (arm 2).

To confirm the sufficiency of the follow-up period for the mice to develop an immune response against the fungal infection, the IFNγ, IL10, WBC, and ALC measures in the control mice (arm 1) were compared with the mice receiving *C. albicans* inoculation with oral placebo (arm 3).

To confirm the sufficiency of the follow-up period for *in vivo* dissemination of *C. albicans*, the fungal burden values in the control mice (arm 1) were compared with the arm receiving *C. albicans* inoculation with oral placebo (arm 3).

To characterize the effect of FTY720 on mice's immunological response to *C. albicans*, the IFNγ, IL10, WBC, and ALC measures in the arms 4,5,6 (i.e., the mice receiving both FTY720 and *C. albicans* inoculation) were once compared with the mice receiving FTY720 with placebo injection (arm 2), and once with the mice receiving *C. albicans* inoculation with oral placebo (arm 3).

In order to characterize the effect of FTY720 on the *in vivo* fungal burden, the liver, kidney, and vaginal fungal burden values in arms 4,5,6 (i.e., the mice receiving both FTY720 and *C. albicans* inoculation*)* were compared with the mice receiving *C. albicans* inoculation with oral placebo (arm 3).

**2.5.3 Software.** The Prism software (version 9.0.2 for MacOS; GraphPad Software LLC.) was used for statistical analysis and graphing.

## 2.6 Reproducibility, safety and ethical considerations

In order to ensure the reproducibility of the results, the Enhancing the QUAlity and Transparency Of health Research (EQUATOR) guidelines [38] were followed to report the study, enabling all researchers across the globe to repeat the experiments, obtain, compare, and validate the results.

The laboratory researchers followed strict regulatory standards in terms of safety. Microbiological Petri dishes were not handled without gloves during any of the experiments. The laboratory researchers wore sterile gowns, protective face-shields and masks, and followed standard hand-washing protocols with povidone iodine before entering and after exiting the laboratory. All of the safety procedures were approved by the regulatory scientific review board who were monitoring the experiments through their surveillance systems.

The animal subjects of this study were kept in standard cages and were not restricted in terms of space, water, and food. All procedures on animal subjects were executed in accordance with national ethical guidelines. This study was approved by the ethics committee of the Islamic Azad University Falavarjan branch (Approval ID: IR.IAU.FALA.REC.1401.010).

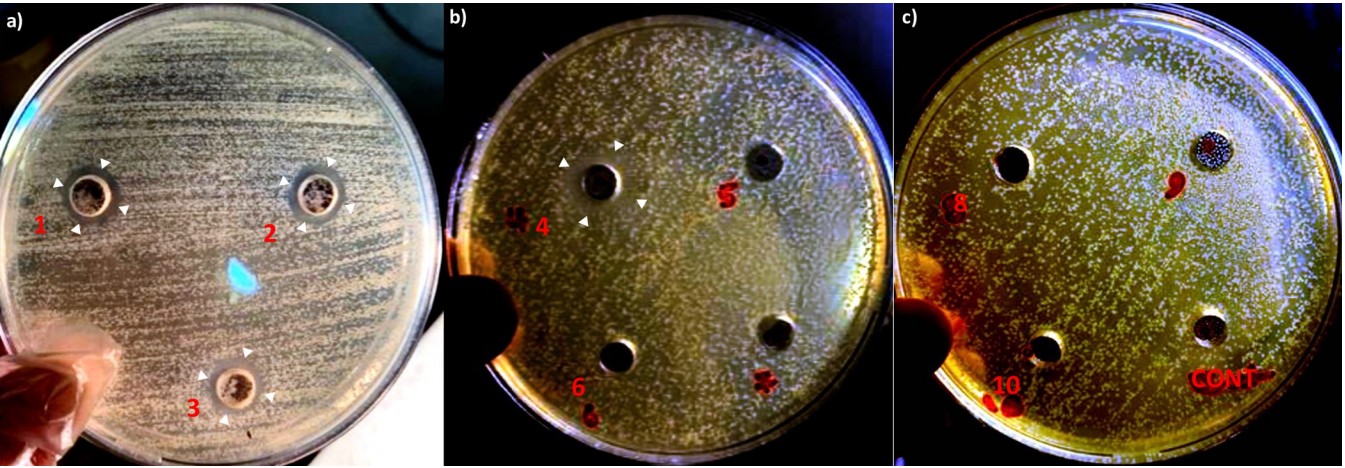

**Fig 2. Final results of the well diffusion assay.** First, standard *C. albicans* preparations (0.5 McFarland, 1–5 x $10^6$ CFU/mL) were spread on SDA plates and eleven sample wells were created. The wells numbered from 1 to 10 in the photographs **(a)**, **(b)**, and **(c)** were then filled with serial concentrations of FTY720 from 1.5 to 0.005 mg/mL respectively. The well labelled as "CONT" in photograph **(c)** was filled with normal saline. Then, the plates were incubated for 30 hours in 37˚C, after which, the results were documented and the depicted photographs were taken. Zones of inhibition (white arrowheads) are visible surrounding the wells numbered from 1 to 4 in the photographs **(a)** and **(b)**, corresponding to serial concentrations of FTY720 from 1.5 to 0.5 mg/mL. Abbreviations: SDA, Sabouraud dextrose agar; CONT, control; CFU, colony forming unit.

## 3. Results

### 3.1 *In vitro* antifungal susceptibility tests

In the well diffusion assay, the lowest concentration of FTY720 for which a visible zone of inhibition surrounded its corresponding well was 0.5 mg/mL (well number 4 in Fig 2B), hence, the probable MIC of FTY720 was estimated to be 0.5 mg/mL or less. As mentioned, we then used the resulted qualitative measures for precise and standard MIC measurement with a broth microdilution assay, in which the $MIC_{99}$ was measured to be 0.25 mg/mL, and the $MIC_{50}$ was measured to be 0.12 mg/mL (Fig 3). These results confirmed that FTY720 comprises antifungal effects *in vitro*.

### 3.2 Overview and validation of the *in vivo* study

The study was conducted per protocol; no unintended events happened during the study. The mice were randomized and allocated to the study arms two days before receiving their oral interventions (T -2). All of the randomized mice were followed up for 23 days (until T +21), except for one mouse in the control arm (arm 1) which died due to an unknown reason in the second day of oral normal saline consumption (T +2, four days after randomization). An investigative dissection of the mouse was conducted to identify the reason of death, but no gross abnormalities were detected. Samples from the normal saline administered to the mouse were cultured, resulting in growth of no microorganisms after one week. Apart from the signs of sepsis in the ones receiving the IP *C. albicans* inoculation, the mice experienced no serious adverse events during their follow-up.

At the end of the study, the immunological profile of the mice treated with FTY720 and injectable placebo (arm 2) significantly differed from the controls (arm 1) (Fig 4A), indicating a sufficient follow-up period for FTY720 to execute its immunomodulatory effects. FTY720 significantly decreased IFNγ, increased IL10, and decreased white blood cell (WBC) and absolute lymphocyte (AL) counts (Fig 4A).

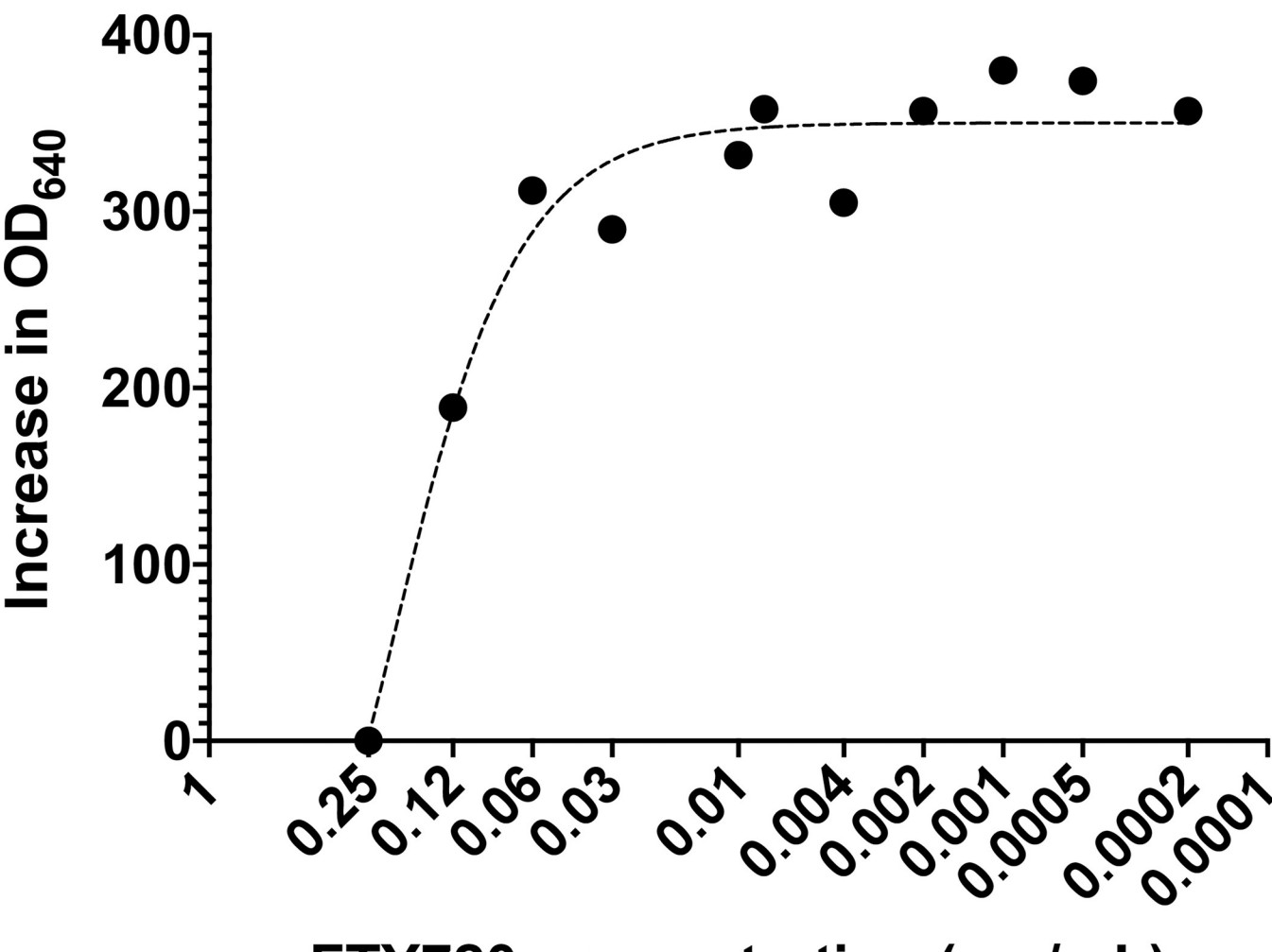

**Fig 3. Results of the broth microdilution assay for determining FTY720's MIC for *C. albicans*.** First, microtubes were filled with standard *C. albicans* preparations in SDB and serial concentrations of FTY720. Then, the background OD's were measured at 640 nm and the microtubes were incubated for 24 hours at 37°C. After that, the OD's were measured again at 640 nm and the results were documented. The measures plotted on the Y-axis are resulted from subtracting the pre-incubation (background) OD's from the post-incubation OD's. The hyphenated line corresponds to the best-fit regression model. Abbreviations: MIC, minimum inhibitory concentration; SDB, Sabouraud dextrose broth; OD, optical density.

The *C. albicans*-infected mice treated with oral normal saline (arm 3) showed significantly-different immunological profiles and fungal burden measures compared to the controls (arm 1) (Fig 4A); this indicated that the follow-up period was sufficient for *in vivo* dissemination of *C. albicans*, and for an antifungal immunological response to develop. Disseminated candidiasis significantly increased IFNγ, WBC, and AL counts but affected IL10 insignificantly (Fig 4A). Cultures of livers, kidneys, and vaginal samples of the mice not receiving IP *C. albicans* inoculation showed no growth of fungi, whereas the cultures from all of the *C. albicans*-injected ones showed growth of *C. albicans*–confirmed with gross visual examination and direct smear microscopy (Figs 5 and 6). In two cultures (one kidney culture from a mouse in arm 3 and one vaginal sample culture from a mouse in arm 6) multiple microorganisms were colonized. As this was highly indicative of procedural contamination, the mentioned cultures were excluded from the analyses.

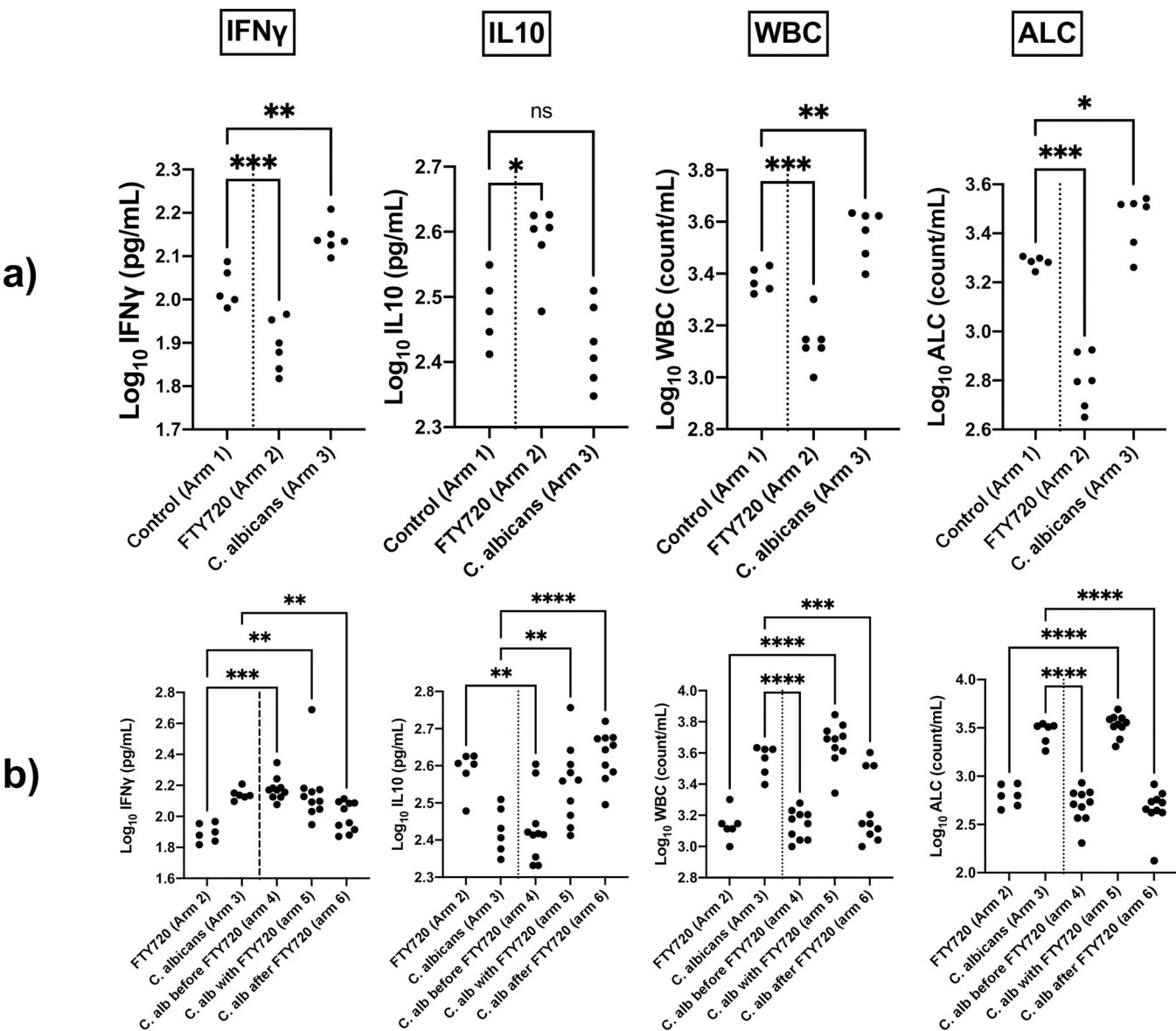

**Fig 4. Immune profile of the mice at the end of the *in vivo* study stratified by study arms.** **(a)** comparison of immune response measures between the control mice (arm 1), the mice receiving FTY720/placebo injection (arm 2), and the ones receiving *C. albicans* inoculation/oral placebo (arm 3). These measures confirm that the follow-up duration of the *in vivo* study was sufficient. **(b)** comparison of immune response measures between the mice receiving either FTY720/placebo injection (arm 2) or *C. albicans* inoculation/oral placebo (arm 3), and the mice receiving both FTY720 and *C. albicans* inoculation (arms 4,5,6); only comparisons with $p<0.05$ are shown. ns: $p>0.05$; *$0.01<p<0.05$; **$0.001<p<0.01$; ***$0.0001<p<0.001$; ****$p<0.0001$. Abbreviations: IFNγ, interferon-gamma; IL10, interleukin-10; WBC, white blood cell; ALC, absolute lymphocyte count; IP, intraperitoneal; C. alb, *Candida albicans*.

### 3.3 Effect of FTY720 on mice's immune response to C. albicans

In order to characterize the effect of FTY720 on the mice's immune response to *C. albicans*, we compared the mice receiving the IP *C. albicans* inoculation before, after, or simultaneously with initiation of FTY720 (arms 4,5,6), with the ones receiving FTY720/injectable placebo or *C. albicans*/oral placebo (arms 2,3) (Fig 1).

Similar to the mice receiving *C. albicans* inoculation/oral placebo (arm 3), the mice receiving *C. albicans* inoculation before or simultaneously with initiation of FTY720 (arms 4,5)

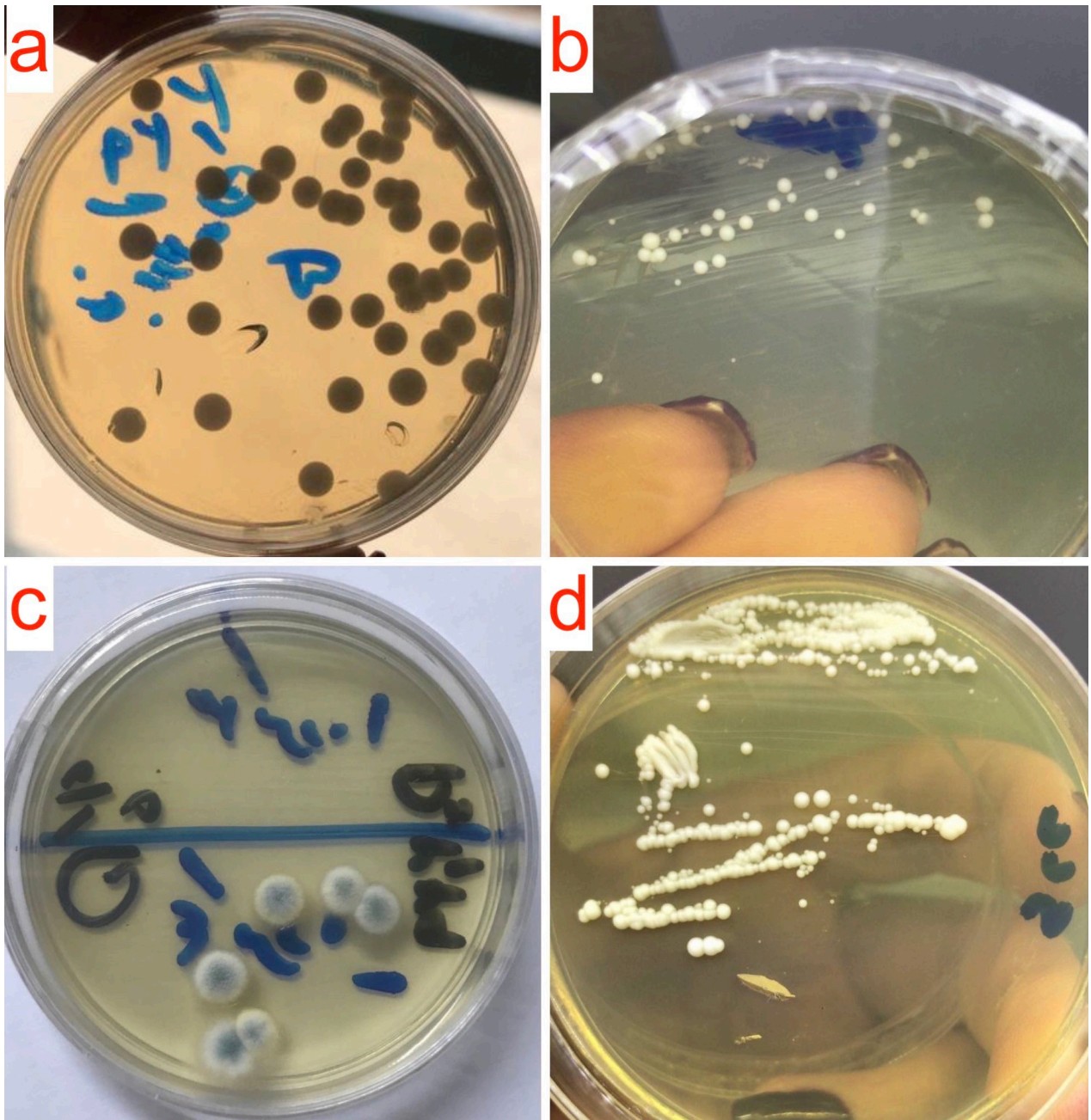

**Fig 5. Final results of four selected mice's organ cultures.** First, the mice's organs were extracted and prepared through sterile procedures and spread on SDA plates. Then, the plates were incubated for 30 hours in 37˚C, after which, the colonies were counted, the results were documented and the depicted photographs were taken. It should be noted that the photographs were taken *after* completion of the experiments and documentation of their results, and that *during* the experiments, none of the microbiological Petri dishes were handled without gloves as seen in these photographs. **(a)** Vaginal sample from a mouse in arm 5; **(b)** Liver sample from a mouse in arm 6; **(c)** Vaginal sample from a mouse in arm 2 (above) and a mouse in arm 3 (below); **(d)** Kidney sample from a mouse in arm 3. Abbreviations: SDA, Sabouraud dextrose agar.

showed significantly increased IFNγ levels compared to the mice receiving FTY720/placebo injection (arm 2) (Fig 4B) (eTable 1 in S1 File). However, the mice receiving *C. albicans* inoculation after two days of FTY720 consumption (in arm 6) showed only insignificantly higher levels of IFNγ compared to the mice receiving FTY720/placebo injection (arm 2); IFNγ levels

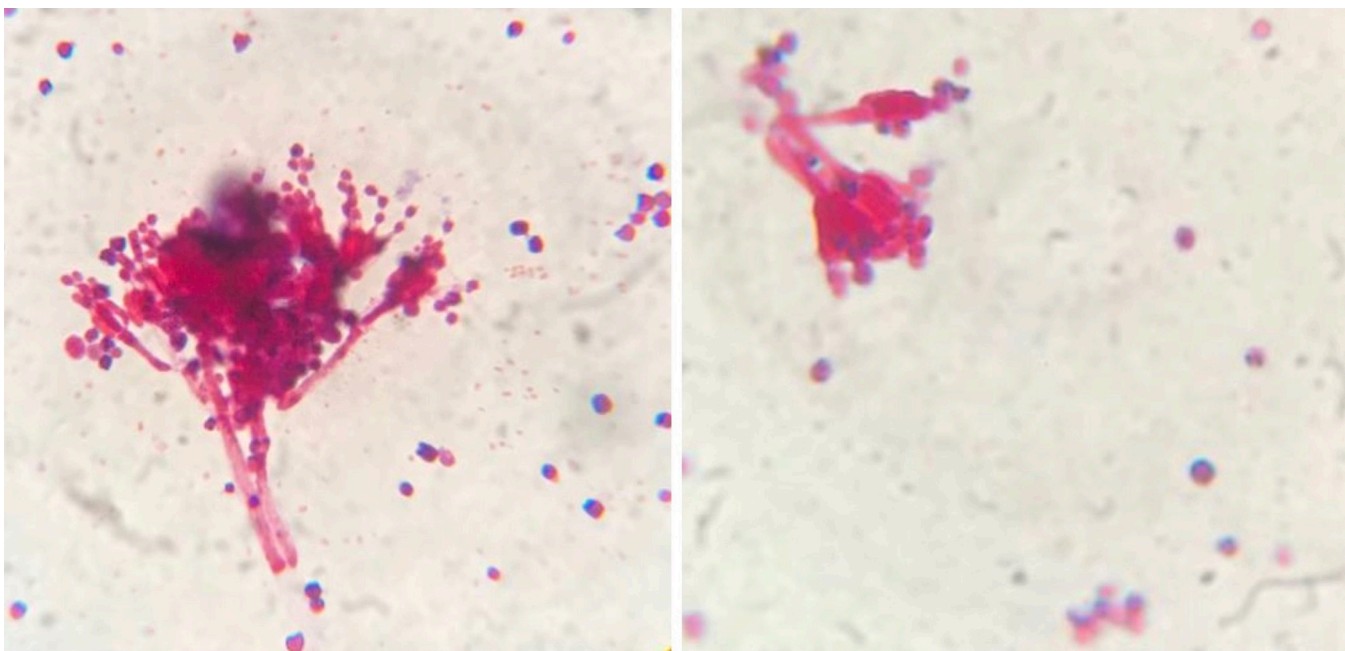

**Fig 6. Micrographs of gram-stained smears of colonies in vaginal sample cultures of two mice in arm 5.** The depicted micrographs were taken from gram-stained slides prepared with the colonies from mice's vaginal sample cultures on SDA plates. *C. albicans* yeasts are seen undergoing filamentation in the depicted micrographs. Abbreviations: SDA, Sabouraud dextrose agar.

were significantly lower in these mice compared to the mice receiving *C. albicans* inoculation/oral placebo (arm 3) (Fig 4B) (eTable 1 in S1 File). In simple words, at the time of fungal inoculation, being on FTY720 for at least two days hindered the IFNγ response of the mice to the disseminating fungal infection, while initiating FTY720 in later timepoints did not.

The mice infected before initiation of FTY720 (arm 4) had significantly lower levels of IL10 compared to the ones receiving FTY720/placebo injection (arm 2), while being comparable to the ones receiving *C. albicans* inoculation/oral placebo (arm3) (Fig 4B) (eTable 1 in S1 File). It could be interpreted that the prior fungal infection hindered the IL10-increasing effect of FTY720. Induction of candidiasis simultaneously with, or later than FTY720 initiation (in arms 5,6) did not affect the IL10-increasing effect of FTY720 as shown in Fig 4B.

As interpreted before, FTY720 significantly decreased, and candidiasis significantly increased WBC and AL counts in mice (Fig 4A). The mice which started taking FTY720 before or after *C. albicans* inoculation (arms 4,6) showed significantly lower WBC and AL counts than the ones receiving *C. albicans* inoculation/oral placebo (arm 2) (Fig 4B). Surprisingly, the mice which initiated FTY720 and were inoculated with *C. albicans* on the same day still showed increased WBC and AL counts–significantly higher than the mice receiving FTY720/injectable placebo (arm 2), and comparable to the ones receiving *C. albicans*/oral placebo (arm 3) (Fig 4B) (eTable 1 in S1 File).

### 3.4 Effect of FTY720 on fungal burden *in vivo*

To demonstrate the antifungal effect of FTY720 *in vivo*, after 21 days of receiving their oral interventions, we cultured the mice's livers, kidneys, and vaginal samples, and compared the resulted fungal burden values between the mice receiving *C. albicans* inoculation/oral placebo (arm 3), and the ones receiving both FTY720 and the *C. albicans* inoculation (arms 4,5,6). The mice initiating FTY720 before, after, or on the same day as *C. albicans* inoculation showed

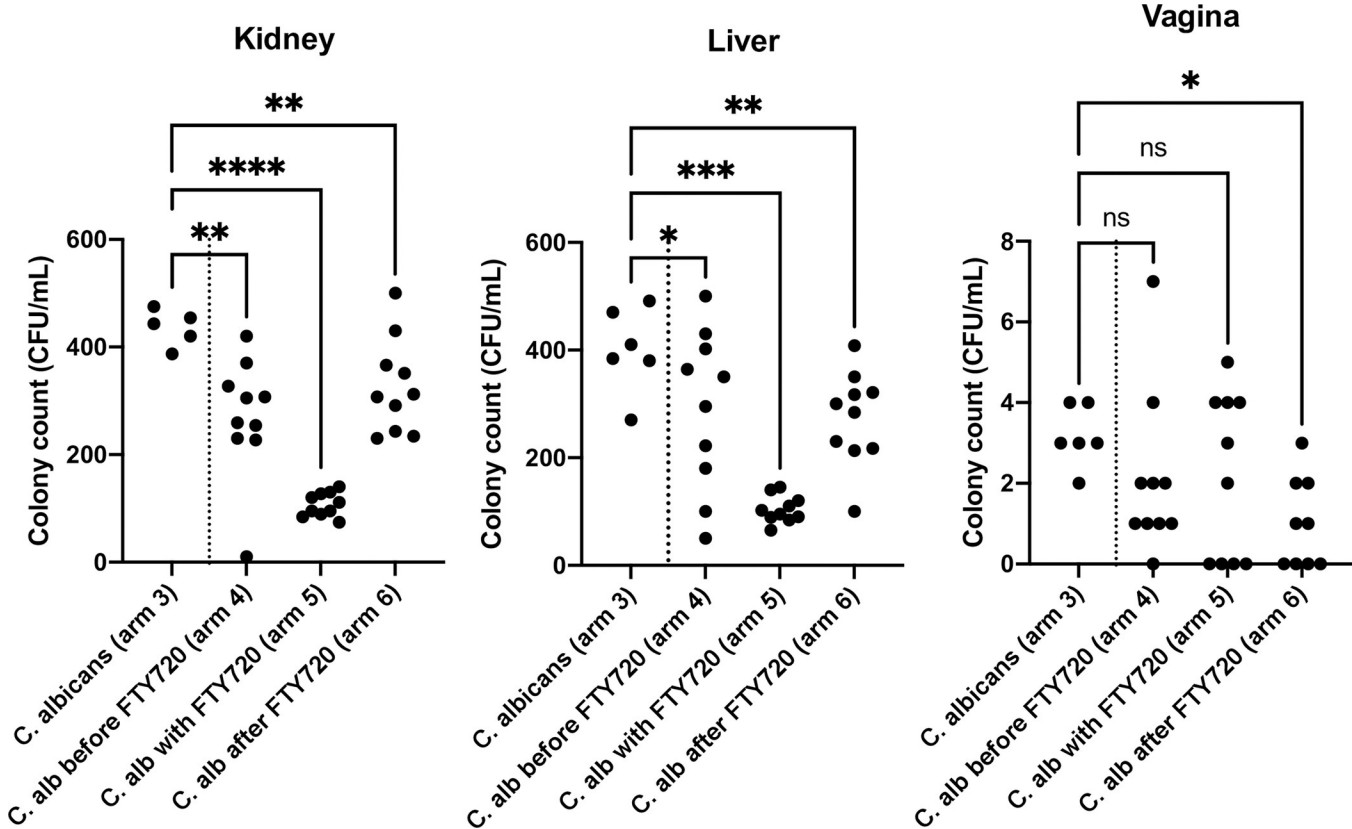

**Fig 7. Colony counts of *C. albicans* in kidneys, livers, and vaginas of the mice inoculated with *C. albicans* at the end of the *in vivo* study, i.e., after 21 days of oral intervention.** ns: $p > 0.05$; *$0.01 < p < 0.05$; **$0.001 < p < 0.01$; ***$0.0001 < p < 0.001$; ****$p < 0.0001$. Abbreviations: C. alb, *Candida albicans*.

significantly lower fungal burden measures in their livers and kidneys (eTable 1 in S1 File) (Fig 7); interestingly, the ones initiating FTY720 and receiving the *C. albicans* inoculation on the same day (arm 5)–which also showed the high AL counts before–had the lowest liver/kidney fungal burdens–significantly lower than all of the other mice in other arms. Regarding the vaginal burdens, only the mice initiating FTY720 before receiving the *C. albicans* inoculation (arm 6) showed lower values than the ones receiving *C. albicans* inoculation/oral placebo (arm 3) (eTable 1 in S1 File) (Fig 7).

## 4. Discussion

Our results demonstrated that FTY720 has antifungal effects against *C. albicans*–both *in vitro* and *in vivo*. In the mice models of disseminated candidiasis, this effect varied based on the chronological relation of FTY720 initiation and *C. albicans* inoculation. Our report is the first describing the *in vivo* antifungal effect of FTY720 as far as we know.

In line with our study, a recent *ex vivo* study by Wei and colleagues pointed to the fungicidal effect of FTY720, which synergized the effect of amphotericin B [39]. Wei et al. showed that this fungicidal effect is present against non-albicans *Candida spp.*, *Saccharomyces cerevisiae*, and *Cryptococcus neoformans* as well as *C. albicans* [39]. They also showed that 0.011 mg/mL of FTY720 affects *C. albicans'* growth rate similar to 0.08 mg/L of amphotericin B [39]. In an attempt to characterize the mechanism of this effect, they demonstrated that FTY720 induces production and hyperaccumulation of reactive oxygen species (ROS) in a dose-

dependent manner. Scavenging the ROS from the samples using N-acetyl cysteine compromised the fungicidal effect of FTY720 significantly, therefore, Wei et al. suggested hyperaccumulation of ROS as the mechanism of FTY720's fungicidal effect [39]. ROS accumulation in fungi induced by unphosphorylated FTY720 has been observed in previous studies as well [15, 40], the upstream processes of which are areas of active investigation.

Administration of FTY720 at 1.25 mg/day in healthy adults–above which has proved unbeneficial for people with MS [7]–results in a maximum blood concentration ($C_{max}$) of 10.2 (±2.7) ng/mL at steady-state [9]; this concentration is 25000 times lower than its $MIC_{99}$ for *C. albicans*. Maximum concentrations of FTY720 in kidney and liver tissues reach approximately 40 times the $C_{max}$ [41]–still much less than its *in vitro* $MIC_{99}$. Therefore, the current FTY720 regimens in people with MS are doubted to directly restrict *C. albicans* infection. In our study *in vivo*, the mice received 0.3 mg/kg/day of FTY720. A single 0.3 mg/kg dose of FTY720 is associated with a maximum liver and kidney concentration of around 1.5 μg/mL [41]. Considering a 10-fold steady-state concentration reached by daily administration, the maximum concentration of FTY720 in liver and kidney reaches 15 μg/mL– 17 times less than its *in vitro* $MIC_{99}$ and 8 times less than its $MIC_{50}$. Despite this fact, we did observe the antifungal effect of FTY720 *in vivo*. Hence, we speculate that FTY720 had a synergistic effect with host's innate immunity against *C. albicans*. The yeast could antagonize oxidative stress up to a specific threshold by inducing antioxidant metabolic pathways [42]. The innate immunity is absent *in vitro*, therefore, higher concentrations of FTY720 are needed to raise the ROS levels up to the threshold intolerable by the yeast. The ROS generation induced by FTY720 *in vivo*, although may not be adequate by itself, is added by the ROS produced by the innate immune cells, attracts more cells to the frontline [43], facilitates phagocytizing, and eases the killing of fungi by the phagocytes [39]. Nevertheless, countless processes are involved in fungal proliferation and expansion *in vivo*; determination of our observations' mechanistic background could be an interesting subject for future studies.

As mentioned in the introduction, some case studies focusing on atypical cryptococcal infections [16, 17] hypothesize that people with MS receiving FTY720 may be generally more susceptible to fungal infections. However, Wei et al. demonstrated that the *ex vivo* fungicidal effect of FTY720 against *Cryptococcus neoformans* is similar to its effect on *C. albicans*, and we demonstrated that the *in vitro* antifungal effect of FTY720 against *C. albicans* is also present *in vivo*. The dose-dependency of FTY720's antifungal effect may be the explanation behind this discrepancy. It could be hypothesized that FTY720's immunosuppressive effect when administered chronically with low doses–as in people with MS–facilitates fungal infections, while its antifungal effects are absent in concentrations reached with those regimens (discussed above).

Another notable observation in our study was the mice with simultaneous *C. albicans* inoculation and initiation of FTY720 (arm 5) had a prominent increase in AL counts, similar to the mice receiving *C. albicans* inoculation/oral placebo (arm 3); however, their kidney and liver samples revealed significantly lower fungal burdens. The decreased fungal burden values in these mice could not merely be explained by the increased AL counts, otherwise, the mice receiving *C. albicans* inoculation/oral placebo (arm 3) who had similar AL counts would have similarly had low fungal burdens in their organs. Further fundamental work is warranted to explain this interesting observation.

## 5. Conclusion

We showed that the immunomodulatory drug FTY720 has an antifungal effect *in vitro*, which is present at even lower concentrations *in vivo*. The facilitation of ROS hyperaccumulation by FTY720 –which assists the innate immune system in creating an antifungal environment–may

explain our observations. Future studies are warranted to evaluate further aspects of repositioning FTY720 for antifungal use in clinical settings.

## 6. Limitations

In order to ensure a sterile and prompt procedure of homogenization and fungal culturing, we regret that we did not perform any sectioning, staining, and direct microscopy of the livers, kidneys, and vagina samples to provide visual confirmation of *C. albicans* colonization *within* the tissues of the mice in treated and control groups. Also, mainly due to limited resources, the *in vitro* effect of FTY720 on important virulence traits of *C. albicans*, including filamentation, adhesion, and biofilm formation was not investigated in any standardized manner. Furthermore, the photographic documentations of the experiments which are provided in this report may not be considered in accordance with research standards; therefore, future studies are advised to replicate our experiments with research-grade photographic documentations to ensure the reproducibility of the results. Other limitations included outcome measurement at a single timepoint rather than several timepoints during the study, usage of limited number of assays, relatively short follow-up period, open-label fashion, not measuring drug concentrations in blood and tissues after administration, not measuring the ROS concentrations *in vitro* or *in vivo*, and other possible unnoticed biases. Further replicative studies are encouraged to add to our knowledge and account for the limitations of our study.

## Supporting information

**S1 File. Contains eTable 1.**
(DOCX)

## Author Contributions

**Conceptualization:** Niloofar Najarzadegan.

**Data curation:** Niloofar Najarzadegan.

**Formal analysis:** Nahad Sedaghat.

**Investigation:** Niloofar Najarzadegan.

**Methodology:** Niloofar Najarzadegan, Nahad Sedaghat.

**Project administration:** Niloofar Najarzadegan.

**Resources:** Niloofar Najarzadegan.

**Supervision:** Mahboobeh Madani, Masoud Etemadifar.

**Validation:** Niloofar Najarzadegan, Mahboobeh Madani.

**Visualization:** Nahad Sedaghat.

**Writing – original draft:** Niloofar Najarzadegan, Nahad Sedaghat.

**Writing – review & editing:** Masoud Etemadifar, Nahad Sedaghat.

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
