## [Decision Letter · Decision Letter 0]

7 Jul 2022

PONE-D-22-15968Immunomodulatory drug fingolimod (FTY720) restricts the growth of opportunistic yeast Candida albicans in vitro and in a mouse candidiasis modelPLOS ONE

Dear Dr Nahad Sedaghat

Thank you for submitting your manuscript to PLOS ONE. After careful consideration, we feel that it has merit but does not fully meet PLOS ONE’s publication criteria as it currently stands. Therefore, we invite you to submit a revised version of the manuscript that addresses the points raised during the review process.

We look forward to receiving your revised manuscript.

Kind regards,

Aijaz Ahmad, Ph.D.

Academic Editor

PLOS ONE

Journal Requirements;

2. In your Methods section, please provide additional information on the animal research and ensure you have included details on : (1) methods of sacrifice (2) methods of anesthesia and/or analgesia, and (3) efforts to alleviate suffering.

Reviewers' comments:

Reviewer's Responses to Questions

**Comments to the Author**

1. Is the manuscript technically sound, and do the data support the conclusions?

Reviewer #1: Yes

Reviewer #2: Yes

Reviewer #3: Partly

2. Has the statistical analysis been performed appropriately and rigorously? 

Reviewer #1: Yes

Reviewer #2: Yes

Reviewer #3: Yes

3. Have the authors made all data underlying the findings in their manuscript fully available?

Reviewer #1: Yes

Reviewer #2: Yes

Reviewer #3: Yes

4. Is the manuscript presented in an intelligible fashion and written in standard English?

Reviewer #1: No

Reviewer #2: Yes

Reviewer #3: No

5. Review Comments to the Author

Reviewer #1: Comments after review:

1. The authors are advised to go through the entire manuscript and correct it for grammatical errors.

2. Authors should mention protocol number approved by Institutional Ethics Committee (IEC) in the manuscript.

3. In in-vitro well diffusion assay, how did the authors manage to load 200ml of sample in the wells. Also, please mention the zone of inhibition values obtained.

4. Please mention the histopathological images of livers, kidneys and vagina obtained after 21 days of treatment to provide visual analysis of Candida colonization between the treated and control groups.

5. Authors have mentioned that Fingolimod restricts the growth of opportunistic yeast Candida albicans in vitro and in a mouse candidiasis model. Have the authors investigated the in vitro effect of this compound on important virulence traits mainly- hydrolytic enzymes, yeast to hyphae transition, adhesion and biofilm formation. If yes, please mention this in manuscript.

Reviewer #2: Overall the necessary experiments were performed in this study. However some structural changes are needed.

1. the in vitro results should precede the in vivo. The figures should follow the same logic.

2. L187-190. Please provide data/figure against this observation.

3. L 106, was it 200 ul of suspension? Figures against the well diffusion assay need to be added.

4. What is the reasoning behind performing the well diffusion assay when the authors performed a more sensitive broth dilution assay?

5. Please stick to the convention of using IFNγ.

6. What is the significance of measuring IFNg, IL10, and WBC, and ALC, What role do they play against infection?

7. If FTY720 has antifungal activity both in vitro and in vivo then what causes fungal infections are reported among people receiving FTY720, mentioned in the introduction?

Reviewer #3: The manuscript entitled “Immunomodulatory drug fingolimod (FTY720) restricts the growth of opportunistic yeast Candida albicans in vitro and in a mouse candidiasis model” is an interesting work done by the authors. However, the manuscript has several flaws starting from the way of writing to the technicality in the method section

1. The introduction lacks information about the severity of Candida albicans in present world and why authors are only targeting Candida albicans and not non-albicans Candida.

2. The method section is very confusing and should be re-written for better understanding of readers.

3. All the methods lack relevant references

4. Line number 106, should be rechecked for the volume, 200 ml is not correct.

5. Line number 108, justification for 30-hour incubation which is not as per standard guidelines.

6. Section 2.3, which labelling technique was used, which program was used and what was the basis for dose determination.

7. Section 2.3, is it possible to withdraw 20 ml of blood from the mice (line 133).

8. Use abbreviations for Sabouraud dextrose agar and other reagents after mentioning at place.

9. Statistical analysis should be re-written for proper understanding

10. Section 3.2, line 194, broth dilution assay should be replaced by microbroth dilution assay.

11. The results need to be discussion with reference to the relevant literature.

12. The manuscript lacks conclusion.

6. PLOS authors have the option to publish the peer review history of their article (what does this mean?). If published, this will include your full peer review and any attached files.

Reviewer #1: **Yes: **SAIEMA AHMEDI Department of Biosciences, Jamia Millia Islamia, New Delhi-110025

Reviewer #2: No

Reviewer #3: No

---

## [Author Response · Author response to Decision Letter 0]

8 Aug 2022

Response to the reviewer comments 

Dear Editor 

Thank you for having our manuscript considered and reviewed. We carefully considered each of the reviewer comments, revised and improved our manuscript accordingly. The changes were tracked in a marked version of the revised manuscript, which is uploaded along the clean version. Below, you can find the reviewer comments, along with our responses below each. Please let us know if any further action is required from us. Thank you in advance for your consideration; we hope to hear from you soon. 

Best regards

Nahad Sedaghat on behalf of the authors 

Reviewer #1

1. “The authors are advised to go through the entire manuscript and correct it for grammatical errors.”

Response: Thank you for reviewing our manuscript and for your constructive suggestions. Per your request, the manuscript was reviewed and edited by a native English scientific writer to ensure proper understanding of the readers.

2. “Authors should mention protocol number approved by Institutional Ethics Committee (IEC) in the manuscript.”

Response: Thank you for noticing. The approval number issued by the ethics committee was added under the “Ethical considerations” section.

3. “In in-vitro well diffusion assay, how did the authors manage to load 200ml of sample in the wells. Also, please mention the zone of inhibition values obtained.”

Response: We loaded 200 “μL” of sample to the wells. Thank you for noticing this error, which is now corrected. Furthermore, it should be mentioned that the well diffusion assay was performed qualitatively only to give us an estimate of the MIC and the required concentrations for the more sensitive and objective broth microdilution assay. In other words, only the qualitative presence of the zones was important to us, as we did not aim to report the MICs based on zone of inhibition cut-offs in well diffusion assays. For the same reason, you could interpret that we did not follow the standard 24-hour incubation protocol to enable MIC estimation based on zone cut-offs. We mistakenly mentioned that the diameters of the zones were measured for MIC determination; this point was corrected. The section on fungal susceptibility assays was reworded while explicitly clarifying and emphasizing the mentioned points to avoid any confusion for the readers. 

4. “Please mention the histopathological images of livers, kidneys and vagina obtained after 21 days of treatment to provide visual analysis of Candida colonization between the treated and control groups.”

Response: In order to ensure a sterile and prompt procedure of homogenization and fungal culturing, we regret that we did not perform any histopathological sectioning of the tissues to provide the requested visual analysis of Candida colonization between the treated and control groups. This point was mentioned in the limitations section. To validate the colonization of C. albicans, we did perform staining and direct microscopy of the colonies after culturing; the available gross and microscopic photographs were added to the supplemental contents.

5. “Authors have mentioned that Fingolimod restricts the growth of opportunistic yeast Candida albicans in vitro and in a mouse candidiasis model. Have the authors investigated the in vitro effect of this compound on important virulence traits mainly- hydrolytic enzymes, yeast to hyphae transition, adhesion and biofilm formation. If yes, please mention this in manuscript.”

Response: Unfortunately, mainly due to limited resources, the in-vitro effect of FTY720 on important virulence traits of Candida albicans was not investigated in any standardized manner. This limitation is now explicitly stated in the limitations section. Thank you once more for dedicating your time to review and improve our manuscript. 

Reviewer #2 

0. “Overall the necessary experiments were performed in this study. However some structural changes are needed.”

Response: Thank you for reviewing our manuscript and providing us with your valuable comments. We hope that we addressed each of your comments sufficiently. 

1. “the in vitro results should precede the in vivo. The figures should follow the same logic.”

Response: Thank you for your suggestion. The figures followed the order they were mentioned in the text in line with the journal requirements. Per your suggestion, we changed the order of the sections 3.1 and 3.2. In the revised manuscript, you could find the in-vitro results in section 3.1, prior to the first description of the in-vivo results in section 3.2. Their corresponding figures also follow the same order in the revised manuscript. 

2. “L187-190. Please provide data/figure against this observation.”

Response: We mentioned that “Cultures from liver, kidney, and vaginal samples from all of the control mice showed no growth of fungi, whereas the samples from all of the C. albicans-injected ones showed growth of C. albicans. Results of two cultures (one kidney sample from an untreated infected mouse and one vaginal sample from a mouse in arm 6) indicated contamination; they were excluded from analyses.” The data showing the fungal colony counts in the cultures of infected mice are interpretable from the diagram presented in Figure 4; raw data is also available upon request (now mentioned in the new “data availability” section). We unfortunately did not follow a standardized manner in photographic documentation of our experiments, nevertheless, some photographs of the culture plates were taken for the record; they are now added to the supplemental contents. Our limitation in providing standardized photographic documentation of the experiments was explicitly stated in the limitations section. The contaminations we mentioned were concluded due to colonization of multiple organisms in the culture plates. This point was added to the manuscript. 

3. “L 106, was it 200 ul of suspension? Figures against the well diffusion assay need to be added.”

Response: Yes, thank you for noticing our error, which is now corrected. In the supplemental contents, we uploaded a series of pictures taken for the record from the well diffusion plates; the zone of inhibition around the wells is visible in those images, but the pictures may not be considered per research standards. As mentioned, we disclosed our limitation in providing standard photographic documentation of the experiments, and also encouraged further replication following the detailed description of methods we provided, while also considering visual validation with guideline-recommended photographic documentations.

4. “What is the reasoning behind performing the well diffusion assay when the authors performed a more sensitive broth dilution assay?”

Response: We performed a qualitative well diffusion assay to have an estimate of the probable MIC measure, in order to determine the concentrations required for the more precise broth microdilution assay. Precise and objective MIC measurement was not intended to be done using zone of inhibition cut-offs in well diffusion assays, which was also the reason we did not follow the standard 24-hour incubation procedure per standard guidelines, and did not measure the MIC based on the zone diameters (as mistakenly stated in the initial draft), but visually assessed the presence of zones. Precise and standard MIC measurement was performed with guideline-recommended procedures using broth microdilution. These points were disclosed explicitly in the revised manuscript. 

5. “Please stick to the convention of using IFNγ.”

Response: Noted. IFNg was changed to IFNγ throughout the manuscript. 

6. “What is the significance of measuring IFNg, IL10, and WBC, and ALC, What role do they play against infection?”

Response: The mentioned measures were obtained as markers of extent and profile of the systemic immune response to the fungal infection. This point was clarified in the revised manuscript.

7. “If FTY720 has antifungal activity both in vitro and in vivo then what causes fungal infections are reported among people receiving FTY720, mentioned in the introduction?”

Response: We discussed and elaborated the mentioned point in the revised manuscript. Thank you once more for your time dedicated to reviewing and contributing to the coherence of our manuscript. 

 

Reviewer #3 

0. “The manuscript entitled “Immunomodulatory drug fingolimod (FTY720) restricts the growth of opportunistic yeast Candida albicans in vitro and in a mouse candidiasis model” is an interesting work done by the authors. However, the manuscript has several flaws starting from the way of writing to the technicality in the method section”

Response: Thank you for dedicating your time to review our manuscript and highlight our errors. We hope we were able to address your comments in a sufficient manner. 

1. “The introduction lacks information about the severity of Candida albicans in present world and why authors are only targeting Candida albicans and not non-albicans Candida.”

Response: Thank you for your suggestion. We elaborated more on the severity of Candida albicans infection in the present world, and the relevance of determining the effect of FTY720 on Candida albicans in our experiment. 

2. “The method section is very confusing and should be re-written for better understanding of readers.”

Response: Thank you for mentioning this important point. To ensure correct understanding of the readers and reproducibility of our work, we have reworded the methods section, this time aiming to choose our words in the least confusing manner, while providing as much detail as possible.

3. “All the methods lack relevant references”

Response: Thank you for noticing. Several relevant references were added to the methods section per your request. Please do not hesitate to let us know more specifically if we are still missing any essential referencing.

4. “Line number 106, should be rechecked for the volume, 200 ml is not correct.”

Response: Indeed. Thank you for noticing. The mentioned error was corrected. 

5. “Line number 108, justification for 30-hour incubation which is not as per standard guidelines.”

Response: Thank you for asking. We did not aim to measure the MIC based on the guideline-recommended zone diameter cut-offs after a 24-hour incubation of the well diffusion plates. We performed a qualitative well diffusion assay in order to estimate the probable MIC and obtain the concentration measures required for the more precise broth microdilution assay. Therefore, as only qualitative visibility of inhibition zones was of value, we incubated the plates for an additional 6 hours. As interpreted, the samples were incubated for 24 hours before precise measurement of MIC, as per EUCAST guideline for the broth microdilution assays. These points were disclosed in the revised manuscript to prevent any confusion.

6. “Section 2.3, which labelling technique was used, which program was used and what was the basis for dose determination.”

Response: The mice were labeled (ie ID-marked) using ear tags. The NumPy package for Python was used for generating a random sequence. For dose determination we decided to obtain a value frequently used in previous studies and proven to have immunomodulatory effects while being safe for the C57BL/6 mice. All of these points were disclosed in the revised manuscript. 

7. “Section 2.3, is it possible to withdraw 20 ml of blood from the mice (line 133).”

Response: Certainly not. We sincerely apologize for and corrected the mentioned error.

8. “Use abbreviations for Sabouraud dextrose agar and other reagents after mentioning at place.”

Response: Thank you for your suggestion. We used abbreviations for the reagents after their first mention.

9. “Statistical analysis should be re-written for proper understanding”

Response: Along the other parts of the methods section, the statistical analysis section was reworded to ensure proper understanding of the readers. 

10. “Section 3.2, line 194, broth dilution assay should be replaced by microbroth dilution assay.”

Response: Thank you for mentioning. We replaced the term “broth dilution” with “microbroth dilution”. 

11. “The results need to be discussion with reference to the relevant literature.”

Response: Per your suggestion, the relevant literature was reviewed and discussed more extensively. 

12. “The manuscript lacks conclusion.”

Response: A “conclusion” section was added at the end of the manuscript. Thank you sincerely for reviewing our manuscript and providing your critical appraisal of our work; your contribution is very much appreciated.

---

## [Decision Letter · Decision Letter 1]

29 Aug 2022

PONE-D-22-15968R1Immunomodulatory drug fingolimod (FTY720) restricts the growth of opportunistic yeast Candida albicans in vitro and in a mouse candidiasis modelPLOS ONE

Dear Dr. Sedaghat

Thank you for submitting your manuscript to PLOS ONE. After careful consideration, we feel that it has merit but does not fully meet PLOS ONE’s publication criteria as it currently stands. Therefore, we invite you to submit a revised version of the manuscript that addresses the points raised during the review process.

We look forward to receiving your revised manuscript.

Kind regards,

Aijaz Ahmad, Ph.D.

Academic Editor

PLOS ONE

Journal Requirements:

Reviewers' comments:

Reviewer's Responses to Questions

**Comments to the Author**

1. If the authors have adequately addressed your comments raised in a previous round of review and you feel that this manuscript is now acceptable for publication, you may indicate that here to bypass the “Comments to the Author” section, enter your conflict of interest statement in the “Confidential to Editor” section, and submit your "Accept" recommendation.

Reviewer #1: All comments have been addressed

Reviewer #2: All comments have been addressed

Reviewer #3: All comments have been addressed

2. Is the manuscript technically sound, and do the data support the conclusions?

Reviewer #1: Yes

Reviewer #2: Yes

Reviewer #3: Yes

3. Has the statistical analysis been performed appropriately and rigorously? 

Reviewer #1: Yes

Reviewer #2: Yes

Reviewer #3: Yes

4. Have the authors made all data underlying the findings in their manuscript fully available?

Reviewer #1: Yes

Reviewer #2: No

Reviewer #3: Yes

5. Is the manuscript presented in an intelligible fashion and written in standard English?

Reviewer #1: No

Reviewer #2: Yes

Reviewer #3: Yes

6. Review Comments to the Author

Reviewer #1: 1. Scientific notation of writing *Candida albicans* is not correct in abstract. Please check. It should be uniform throughout the manuscript.

2. Please incorporate the images of well diffusion assay in the manuscript.

3. Authors should keep space between values and their units throughout the manuscript and

used SI units.

4. Authors are advised to incorporate images to get better picture of their results.

5. Please incorporate toxicity related studies in the manuscript. If already published by others cite that reference in the manuscript.

Reviewer #2: 

The authors have answered have all the questions raised by me. However, as the authors mentioned in the article that there are some significant limitations to the reproducibility of the experiments. The supplementary figures are also not as per scientific standards, especially handling microbiological Petri dishes without gloves is a serious concern. There are still a few typos in the manuscript (eg. Line 63 homo-, IFNg instead of γ in the etable etc.). Therefore, I recommend to resubmit the article with proper figures.

Reviewer #3: (No Response)

7. PLOS authors have the option to publish the peer review history of their article (what does this mean?). If published, this will include your full peer review and any attached files.

Reviewer #1: No

Reviewer #2: No

Reviewer #3: No

---

## [Author Response · Author response to Decision Letter 1]

28 Sep 2022

Reviewer #1: 

1. Scientific notation of writing Candida albicans is not correct in abstract. Please check. It should be uniform throughout the manuscript.

Response: Thank you for noticing. We uniformly changed all the instances in the abstract to “Candida albicans”.

2. Please incorporate the images of well diffusion assay in the manuscript.

Response: Thank you for your suggestion. All eFigures were moved to the main manuscript. 

3. Authors should keep space between values and their units throughout the manuscript and used SI units.

Response: Thank you for mentioning. We added the required space between the values and units. About using SI units, as far as we checked, we did not use non-SI units anywhere in the manuscript. In case we overlooked an instance of using any non-SI unit, we would appreciate if you could point to the line, so we could correct that instance. 

4. Authors are advised to incorporate images to get better picture of their results.

Response: Thank you for your suggestion. As mentioned, we moved all the eFigures to the main manuscript. We unfortunately have a limitation in presenting further images apart from what is already presented, as the experiments are no longer in progress. We mentioned this point in the limitations section and advised future studies to account for this limitation. 

5. Please incorporate toxicity related studies in the manuscript. If already published by others cite that reference in the manuscript.

Response: Thank you for your attention. We added and discussed toxicity related studies as well in the revised manuscript. We hope the changes and improvements satisfy your concerns. 

 

Reviewer #2: 

0. The authors have answered have all the questions raised by me.

Response: We are pleased to hear that. Thank you for dedicating your time to our manuscript. 

1. However, as the authors mentioned in the article that there are some significant limitations to the reproducibility of the experiments.

Response: Thank you for mentioning your concern about the replicability of the study and reproducibility of the results. In case your concern is due to the lack of standard photographic documentation of the experiments and results, we made the following efforts to account for this limitation and make our report of merit for publication. First of all, we described and disclosed all of the experiments in full detail and in complete accordance with the EQUATOR guidelines, in order to allow their complete repetition by any scientist across the globe. Therefore, replication of the experiments and reproduction of the results would be possible for any team concerned about the lack of proper photographic documentations in the present report. Additionally, all of the experiments were performed in compliance with strict regulations and were being monitored through surveillance systems by independent scientific, safety, and ethical review boards, consisting of anonymous nationally-certified scientists. Finally, we shared and deposited our data for all researchers across the globe to review and validate. We would also welcome any further suggestions to improve our work. 

2. The supplementary figures are also not as per scientific standards. 

Response: Thank you for mentioning your concern. We completely agree that the presented figures are not in compliance with scientific standards, as we unfortunately, did not intend to keep scientific photographic documentations at the beginning. The reason was that in presence of strict regulations and standard guidelines, we erroneously considered photographic documentations to be optional. Now, as the experiments are no longer in process, we are unfortunately limited in terms of presenting standard figures. However, describing the experiments with full detail allows their repetition by any researcher concerned about the reproducibility of the results. In the limitations section, we explicitly stated this point, and strongly encouraged further replication of the experiments using the detailed description of the methods we provided. We further stated that due to our limitation in providing standard figures, our conclusions are subject to reproduction of the results by future studies with standard photographic documentations.

3. especially handling microbiological Petri dishes without gloves is a serious concern. 

Response: Thank you for your punctiliousness. We completely acknowledge your concern. We should mention that the photograph you are referring to was taken after completion of the experiments and documentation of their results. During the experiments, the laboratory researchers followed strict regulatory standards in terms of their safety; microbiological Petri dishes were not handled without gloves during any of the experiments, and the laboratory researchers wore sterile gowns, protective face-shields and masks, as well as following standard hand-washing protocols with povidone iodine solution before entering the laboratory. This has been evidenced by absence of biological and non-biological contamination in all of the Petri dishes except for one, and by the approval of the safety procedures by the regulatory scientific review board. These points were clarified in the revised manuscript and in the legend of the mentioned figure to appreciate any concern. 

4. There are still a few typos in the manuscript (eg. Line 63 homo-, IFNg instead of γ in the etable etc.). 

Response: We apologize for overseeing the mentioned typos in our reviews. The Latin phrase “in homo” means “in man” in English; we exchanged “in homo” with the standard English term “in humans” to avoid any confusion. IFNg was changed to IFNγ in the supplementary table as well. Furthermore, we reviewed the revised manuscript several more times to ensure proper usage of language for the readers. We would certainly appreciate any further suggestions for improving our report. 

5. Therefore, I recommend to resubmit the article with proper figures. 

Response: Thank you for your recommendation. As mentioned previously, the experiments are no longer in progress, therefore, we are unfortunately limited in presenting standard photographic documentations. We truly hope our explicit and detailed disclosure of the experimental procedures according to the EQUATOR guidelines and encouragement of future replicative studies could at least, partially resolve your concerns about the reproducibility of the results. We are also, hoping that the publication of our study, if considered of merit based on novelty and relevance, leads to conduction of further studies on the subject accounting for our limitations (including provision of standard figures) and validating the results. Once again, we would like to appreciate your punctiliousness and explicitly in critically appraising our study, and thank you for your time and consideration.

---

## [Decision Letter · Decision Letter 2]

26 Oct 2022

PONE-D-22-15968R2Immunomodulatory drug fingolimod (FTY720) restricts the growth of opportunistic yeast Candida albicans in vitro and in a mouse candidiasis modelPLOS ONE

Dear Dr. Nahad Sedaghat

Thank you for submitting your manuscript to PLOS ONE. After careful consideration, we feel that it has merit but does not fully meet PLOS ONE’s publication criteria as it currently stands. Therefore, we invite you to submit a revised version of the manuscript that addresses the points raised during the review process.

We look forward to receiving your revised manuscript.

Kind regards,

Aijaz Ahmad, Ph.D.

Academic Editor

PLOS ONE

Journal Requirements:

Additional Editor Comments:

Reviewer 1:

1. The annotation of figures is not explicit enough, for example in the legend of figure 2 and 5, no information is provided for the images such as what type of images these are, how processed and the conditions. Please mention.

2. Quality of all disc/plates images is not good. Zone of inhibition is not visible in Figure 2 b and c. Better to crop the image, put it on black or dark background then take the image.

3. Authors are advised to thoroughly check the manuscript for grammatical errors and formatting.

Reviewer 2:

Accept

Reviewer 3:

The manuscript need minor changes before it gets accepted for publication.

I would urge the authors to send their paper for English editing because it has significant grammatical errors that make it difficult to read the work as a whole. There are also several errors that might have been corrected with ease. The examples I provided, which can be found in the mark-up document, are by no means exhaustive.

Green colour- revise whole sentence

Yellow colour - check for the consistency throughout the manuscript

Reviewers' comments:

Reviewer's Responses to Questions

**Comments to the Author**

1. If the authors have adequately addressed your comments raised in a previous round of review and you feel that this manuscript is now acceptable for publication, you may indicate that here to bypass the “Comments to the Author” section, enter your conflict of interest statement in the “Confidential to Editor” section, and submit your "Accept" recommendation.

Reviewer #1: All comments have been addressed

Reviewer #2: All comments have been addressed

Reviewer #3: All comments have been addressed

2. Is the manuscript technically sound, and do the data support the conclusions?

Reviewer #1: Yes

Reviewer #2: Yes

Reviewer #3: Yes

3. Has the statistical analysis been performed appropriately and rigorously? 

Reviewer #1: Yes

Reviewer #2: Yes

Reviewer #3: Yes

4. Have the authors made all data underlying the findings in their manuscript fully available?

Reviewer #1: Yes

Reviewer #2: Yes

Reviewer #3: Yes

5. Is the manuscript presented in an intelligible fashion and written in standard English?

Reviewer #1: No

Reviewer #2: Yes

Reviewer #3: No

6. Review Comments to the Author

Reviewer #1: 1. The annotation of figures is not explicit enough, for example in the legend of figure 2 and 5, no information is provided for the images such as what type of images these are, how processed and the conditions. Please mention.

2. Quality of all disc/plates images is not good. Zone of inhibition is not visible in Figure 2 b and c. Better to crop the image, put it on black or dark background then take the image.

3. Authors are advised to thoroughly check the manuscript for grammatical errors and formatting.

Reviewer #2: The authors have responded to all my concerns. However, it is always a good laboratory practice to properly document all the data and store it properly.

Reviewer #3: The manuscript need minor changes before it gets accepted for publication.

I would urge the authors to send their paper for English editing because it has significant grammatical errors that make it difficult to read the work as a whole. There are also several errors that might have been corrected with ease. The examples I provided, which can be found in the mark-up document, are by no means exhaustive.

Green colour- revise whole sentence

Yellow colour - check for the consistency throughout the manuscript

7. PLOS authors have the option to publish the peer review history of their article (what does this mean?). If published, this will include your full peer review and any attached files.

Reviewer #1: No

Reviewer #2: No

Reviewer #3: No

---

## [Author Response · Author response to Decision Letter 2]

9 Nov 2022

Reviewer #1: 

1. “The annotation of figures is not explicit enough, for example in the legend of figure 2 and 5, no information is provided for the images such as what type of images these are, how processed and the conditions. Please mention.”

Response: Thank you for your notice. Further information including but not limited to the points you mentioned were added to the figure legends to ensure adequate understanding of the readers. 

2. “Quality of all disc/plates images is not good. Zone of inhibition is not visible in Figure 2 b and c. Better to crop the image, put it on black or dark background then take the image.”

Response: Thank you sincerely for your concern and for your suggestion. Following your suggestion and for the zones of inhibition to be visible for the readers, backgrounds were darkened in Figure 2 photographs (showing the results of the well diffusion antifungal susceptibility assay) and the zones of inhibition were marked with white arrowheads. Please note that, no zone of inhibition was present around wells number 5 to 10 which are depicted in Figure 2 b and c, and that no “discs” were used on the plates, but wells were created in the medium and filled with drug suspensions. 

3. “Authors are advised to thoroughly check the manuscript for grammatical errors and formatting. “

Response: Thank you for once again for your suggestion. The manuscript was checked by a native English scientific copyeditor to ensure proper usage of language and formatting. 

Reviewer #3: 

0. “The manuscript need minor changes before it gets accepted for publication. I would urge the authors to send their paper for English editing because it has significant grammatical errors that make it difficult to read the work as a whole. There are also several errors that might have been corrected with ease. The examples I provided, which can be found in the mark-up document, are by no means exhaustive. Green colour- revise whole sentence Yellow colour - check for the consistency throughout the manuscript”

Response: We would like to sincerely thank you for providing us with the mark-up document, which indeed, alleviated the process of further improving our language. All of the points in the mark-up document were addressed in our revision. Also, the manuscript was reviewed by a native English scientific copyeditor to ensure proper usage of syntax and formatting.

---

## [Editor Report · Decision Letter 3]

17 Nov 2022

Immunomodulatory drug fingolimod (FTY720) restricts the growth of opportunistic yeast Candida albicans in vitro and in a mouse candidiasis model

PONE-D-22-15968R3

Dear Dr. Nahad Sedaghat 

We’re pleased to inform you that your manuscript has been judged scientifically suitable for publication and will be formally accepted for publication once it meets all outstanding technical requirements.

Kind regards,

Aijaz Ahmad, Ph.D.

Academic Editor

PLOS ONE

Additional Editor Comments (optional):

Accept
---

## [Editor Report · Acceptance letter]

28 Nov 2022

PONE-D-22-15968R3 

Immunomodulatory drug fingolimod (FTY720) restricts the growth of opportunistic yeast <i>Candida albicans in vitro<i> and in a mouse candidiasis model 

Dear Dr. Madani:

I'm pleased to inform you that your manuscript has been deemed suitable for publication in PLOS ONE. Congratulations! Your manuscript is now with our production department. 

Kind regards, 

on behalf of

Dr. Aijaz Ahmad 

Academic Editor

PLOS ONE